# Similarities and Differences between Danish and American Physicians' Religious Characteristics and Clinical Communication: Two Cross-Sectional Surveys

Christian B. van Randwijk [1], Tobias Opsahl [2], Elisabeth Assing Hvidt [3], Tobias Kvist Stripp [3], Lars Bjerrum [4], Jørn Herrstedt [5], Jens Søndergaard [3] and Niels Christian Hvidt [3,*]

[1] Cogere Consulting, University of Copenhagen, 1165 Copenhagen, Denmark; info@cogere.dk
[2] Sano, 4230 Skælskør, Denmark; to.opsahl@gmail.com
[3] Research Unit of General Practice, University of Southern Denmark, 5000 Odense, Denmark; ehvidt@health.sdu.dk (E.A.H.); tkstripp@health.sdu.dk (T.K.S.); jsoendergaard@health.sdu.dk (J.S.)
[4] Section of General Practice, University of Copenhagen, 1014 Copenhagen, Denmark; lbjerrum@sund.ku.dk
[5] Department of Clinical Oncology and Palliative Care, Zealand University Hospital Roskilde, 4000 Roskilde, Denmark; jherr@regionsjaelland.dk
* Correspondence: nchvidt@health.sdu.dk; Tel.: +45-40717755

**Abstract:** Many physicians remain reticent to initiate or partake in discussions about their patients' religious and spiritual needs during the clinical encounter. Reasons for this may be insufficient time, capacity, education or training but may also be a product of variance in physicians' own religious or spiritual characteristics. The aim of this paper was to compare American and Danish physicians' religious characteristics, and to explore and compare American and Danish physicians' attitudes towards, and practices of, integrating religiosity and spirituality in the clinical encounter. We included data from two cross-sectional surveys: an American survey conducted in 2002 ($n = 2000$) and a Danish survey conducted in 2012 ($n = 1485$) to test four hypotheses. American physicians were significantly more religious, they more frequently inquired about religious or spiritual issues in the clinical encounter and they found it more appropriate to discuss religious or spiritual issues if the patients brought it up when compared to Danish physicians. A weak to moderate positive correlation between level of religiosity and frequency of inquiring about religious and spiritual issues were found in both populations. The findings are discussed in relation to the clinical importance of ensuring that health care practices stay patient centered. The findings may especially be relevant to consider in increasingly ethnically and culturally diverse contexts.

**Keywords:** medical ethics; rehabilitation medicine; palliative care; public health; spirituality; communication

## 1. Introduction

A substantial amount of research has shown the significant role of religiosity and spirituality in many patients' coping and adjustment strategies across different diseases and cultural settings (Koenig 2012). In contrast, the majority of physicians remain reticent to initiate or partake in discussions during the clinical encounter about their patients' religious and spiritual (R/S) needs, resources and practices (Curlin et al. 2006; Hvidt et al. 2016; Kørup et al. 2016). One reason for this may be insufficient organizational resources, such as time and capacity, or lack of education and training in how to address and communicate about existential, spiritual and religious needs of the patient as part of modern, person-centered care (Assing Hvidt et al. 2018; Balboni et al. 2014; Carr 2010; Curlin et al. 2006; McCauley et al. 2005). Another reason may be that some of the variance in disposition towards addressing R/S issues in the clinical encounter is motivated by the physicians' own R/S characteristics (Curlin et al. 2006; Lee et al. 2014).

A growing body of research has investigated the relationship between physician values (such as religious and atheistic values) and the inclination to integrate R/S aspects into patient care. This research shows that physicians who characterize themselves as religious or spiritual are more likely to address R/S issues in the clinical encounter than their atheistic or agnostic colleagues (Curlin et al. 2006, 2007). Most of the research in the relationship between physician values and attitudes towards inclusion of R/S aspects into health care is concerned with differences within particular national populations (Al-Yousefi 2012; Curlin et al. 2006; Lee et al. 2014). To our knowledge, no comparative research on the relationship between physician R/S characteristics and their attitudes towards, and practices of, integrating R/S in health care has been conducted in Western settings. Investigating this issue cross-culturally is central for the understanding of how, and to what degree, medical practice in different cultural settings, but within comparable professional environments, may be influenced by individual and broader cultural and religious characteristics. A comparative analysis could thus contribute to the future development of generalizable theories of the influence of religion and culture on medical practice.

The aim of this paper was therefore to investigate differences and similarities in American and Danish physicians' religious characteristics, and to explore and compare American and Danish physicians' attitudes towards, and practices of, integrating R/S in the clinical encounter.

We tested four hypotheses. Firstly, we expect both American and Danish physicians to be open to discuss R/S issues if patients bring them up, thus respecting the core medical competency of patient-centered care put forth by the World Health Organization (WHO 2005). Secondly, because Americans in general are shown to be more religious than Danes (Andersen et al. 2008; Zuckerman 2008), we expect American physicians to score higher on measures of R/S characteristics than Danish physicians. Thirdly, because research has shown that religious physicians inquire about R/S issues more often than less religious or non-religious physicians (Curlin et al. 2006, 2007; van Randwijk et al. 2019), we expect American physicians to inquire more often about these issues compared to Danish physicians. Finally, we expect that the more time physicians spend on inquiring about R/S issues, the more they feel that the amount of time spent is appropriate, and conversely (Stark and Bainbridge 1985; Stoltz 2008).

## 2. Materials and Methods

### 2.1. Participants

We included data from two cross-sectional surveys: an American survey conducted in 2002 (Curlin et al. 2005) and a Danish survey conducted in 2012 (van Randwijk et al. 2019).

The American questionnaire was sent to 2000 physicians chosen from the American Medical Association Physician Masterfile via stratified random sampling. 1142 American physicians responded (26.3% female), with a median age of 49 years, ranging from 27–65. The religious affiliation of the American physicians was as follows: Christian (56.5%), Jewish (16.1%), no affiliation (10.1%) other (7.0%), Hindu (4.7%), Muslim (2.9), Mormon (1.5%), and Buddhist (1.1%). Participants received up to 3 separate mailings and were offered 20$ in the third mailing.

The Danish questionnaire was mailed to 1485 physicians in the Region of Southern Denmark. The physicians were chosen randomly from a database of The National Board of Health (NBH). 911 Danish physicians responded (42.5% female), with a median age of 49 years, ranging from 26–75. The religious affiliations of the Danish physicians were as follows: Christian (76.2%), no affiliation (20.5%), other (1.9%), Muslim (0.7%), Buddhist (0.2%), and Hindu (0.1%). GPs were offered financial compensation for the time spent filling out the questionnaire (256DKK, approximately equivalent to 35USD), as is common practice in surveys among GPs in Denmark.

### 2.2. Measures

This study was based on the Network for Research on Spirituality and Health (NERSH) data pool which contains questionnaire data on health care practitioners' beliefs, values and practices, gathered across eleven studies conducted by research teams in nine different countries. Details on the development and description of the data pool have been described elsewhere (Hvidt et al. 2016; Kørup et al. 2016). In regard to data on the American physicians, additional details of the methods employed for questionnaire development and data gathering in America have been described elsewhere (Curlin et al. 2005). The same applies to additional details on the Danish questionnaire, sample and data (van Randwijk et al. 2019). The validation process of the questionnaire has been described elsewhere (van Randwijk et al. 2019).

### 2.3. Questionnaires

American and Danish physicians were compared on four issues.

1. R/S characteristics. In order to measure the physicians' R/S characteristics, a factor based on four items was employed: (1) "To what extent do you consider yourself a religious person?"; (2) "To what extent do you agree with the following statement? My religious beliefs influence my practice of medicine"; (3) "To what extent do you agree with the following statement? I try hard to carry my religious beliefs over into all my other dealings in life."; and finally (4) "To what extent do you agree with the following statement? My whole approach to life is based on my religion." The selected items of the factor are based on the Duke University Religion Index (for discussion of the factor, see 4. *Discussion* below) (Koenig et al. 2010). The factor ranged from 4 points to 16, with a score of 4 being the least religious and 16 being the most religious.

2. Frequency of inquiry about R/S issues in clinical situations containing existential topics. Because no single item concerning the frequency of inquiry about R/S was present in both questionnaires, a new measure had to be developed. Using principal-component factor analysis (PCF), one primary factor was identified. It consisted of 4 items: "In the following clinical situations, how often do you inquire about religious/spiritual issues? When a patient presents with . . . (1) faces a frightening diagnosis or crisis, (2) faces the end of life, (3) suffers from anxiety or depression, (4) faces an ethical quandary". The scale showed an overall Cronbach's alpha of 0.92 (N = 1671) with factor loadings of 0.9317, 0.9018, 0.8865, and 0.8839. The primary factor had an Eigenvalue of 3.25 and the screen plot strongly indicated that opting for the one-factor solution was optimal (the difference between primary and secondary factor was 2.93). The scale ranged from 0 to 16 with the score of 0 being never inquiring about R/S issues and 16 being always inquiring about R/S issues in the described situations.

3. Appropriateness of amount of time spent inquiring about R/S issues. In order to investigate differences in American and Danish physicians' view on whether the time they spent inquiring about R/S issues was appropriate, the following item was employed: "Overall, do you think the amount of time you spend inquiring about R/S issues is . . . ", with the following possible answers: too much, too little, and the right amount.

4. Appropriateness of discussing R/S issues if patients bring it up. In order to investigate the hypothesis on the physicians' propensity to discuss R/S issues, one item was employed in the analysis: "In general, is it appropriate or inappropriate for a physician to discuss R/S issues when a patient brings them up?", with the following possible answers: Always appropriate, usually appropriate, usually inappropriate, and always inappropriate. Higher values indicated considering it more inappropriate.

### 2.4. Statistical Analysis

Descriptive statistics, tables, figures and statistical analyses were conducted in Stata, version 13.1. Associations between categorical variables were explored using Pearson's $\chi^2$. Differences between groups on numerical variables were tested with Welch *t*-test. The test was chosen upon finding that variance was unequal for all relevant variables as were the

shapes of the distributions (Delacre et al. 2017; Fagerland and Sandvik 2009). Measures of effect (Cramérs V and Cohen's d) were employed to investigate whether associations or differences between variables were meaningful apart from being statistically significant. Pearson's $\chi^2$ was employed to test the association between nationality and stance towards the appropriateness of the time the physicians spend on inquiring about R/S issues, which were both categorical variables. Independent samples t' tests were used to compare differences in scores of religiosity for American and Danish physicians, to investigate the differences in frequency of inquiring about R/S issues in the clinical encounter between American and Danish physicians, and to investigate the differences in American and Danish physicians' rating of appropriateness of discussing R/S issues if patients brought them up. Associations between; religiosity and frequency of inquiring about R/S issues for each country; and attitude towards appropriateness of inquiring about R/S issues if patients bring them up and frequency of inquiring about R/S issues were investigated using Spearman's correlation.

### 3. Results

#### 3.1. R/S Characteristics

Level of religiosity measured through the religiosity factor mentioned above differed significantly between the two groups, according to Welch t' test, t(1740.03) = 24.23, $p < 0.001$. On average, American physicians were more religious (M = 10.01, SD = 3.27), compared to Danish physicians (M = 6.49, SD = 2.95). The magnitude of the difference in the means (mean difference: 3.53 95% CI = 3.24–3.81) was large (Cohen's d = 1.12, 95% CI = 1.02–1.22). The distributions of scores are provided in Figure 1.

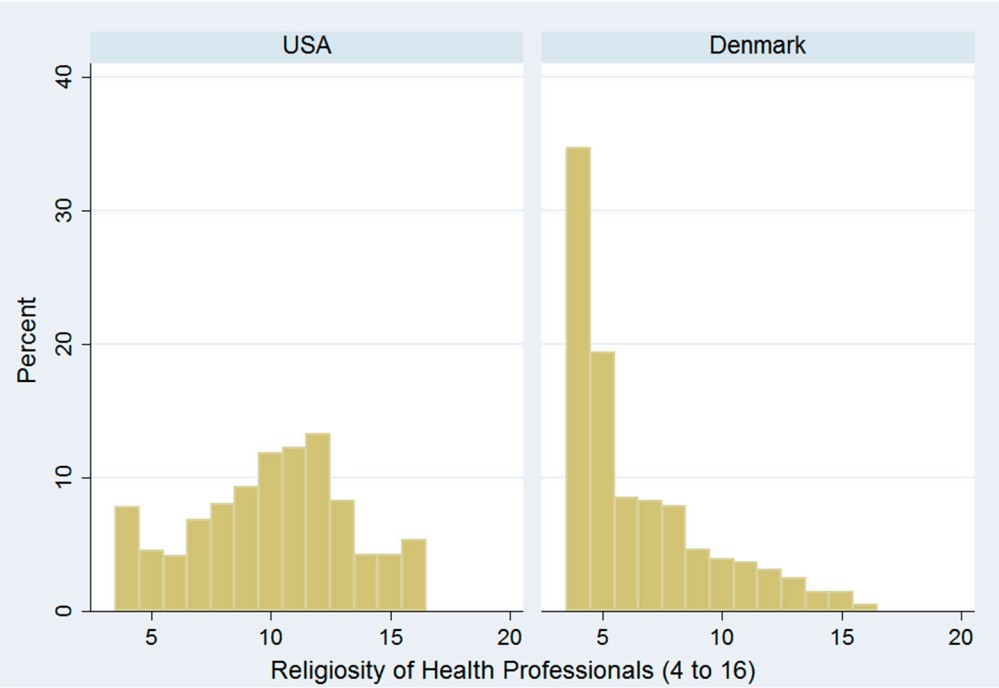

**Figure 1.** Bar chart illustrating the distribution of American and Danish physicians' religiosity. Level of religiosity is measured employing a factor composed of four items. The scale ranges from 4–16, with 4 being the least religious and 16 being the most religious.

#### 3.2. Frequency of Inquiring about R/S Issues

The frequency of inquiring about R/S issues was significantly different between the groups, according to Welch t' test, t(669.65) = 5.2, $p < 0.001$. On average, American physicians more frequently inquired about R/S issues (M = 7.29, SD = 4.41) compared to Danish

physicians (M = 4.32, SD = 3.57). The magnitude of the difference in the means (mean difference: 2.97, 95% CI = 2.59–3.35) was medium to large (Cohen's d = 0.73, 95% CI = 0.63–0.83).

### 3.3. Appropriateness of the Time Spent Inquiring about R/S Issues

In order to investigate the association between nationality and stance towards the appropriateness of the time the physicians spend on inquiring about R/S issues in the clinical encounter, a 2 (nationality) by 3 (appropriateness of amount of time spent) table was constructed and $\chi^2$ analysis was conducted. This revealed that Americans were more likely to report inquiring about R/S issues "too little" than Danish physicians. 37.8% of American physicians reported that they spend too little time inquiring about R/S issues. The same applied to 16.3% of Danish physicians. The association between nationality and stance towards appropriateness of the time they spend discussing R/S issues was significant with a small to medium strength (Pearson's $\chi^2$ (2, N = 1695) = 85.90, $p < 0.001$, Cramér's V = 0.23).

### 3.4. Appropriateness of Discussing R/S Issues If Patients Bring It Up

Significant differences in responses between the two groups were identified, according to Welch t' test, t(1484.46) = −23.87, $p < 0.001$. On average, American physicians deemed it more appropriate to discuss R/S issues if patients bring it up (M = 1.72, SD = 0.62) compared to Danish physicians (M = 2.51, SD = 0.77) (a higher score indicated considering it more inappropriate). The magnitude of the difference in means (mean difference = −0.79, 95% CI = −0.85–−0.72) was large (Cohen's d = −1.14, 95% CI = −1.24–−1.05).

### 3.5. Association between Religiosity and Frequency of Inquiring about R/S Issues

For the American physicians, a statistical significant, but weak to moderate positive correlation between level of religiosity and frequency of inquiring R/S issues was identified ($r_s = 0.39$, $p < 0.001$). In the Danish sample, a statistical significant, weak positive correlation was also revealed ($r_s = 0.29$, $p < 0.001$).

### 3.6. Association between Attitude towards Appropriateness of Inquiring about R/S Issues If Patients Bring Them Up and Frequency of Inquiring

For the sample as a whole, a statistically significant, moderate negative correlation between attitudes towards inquiring about R/S issues, and self-reported frequency of inquiring was identified ($r_s = −0.49$, $p < 0.001$).

## 4. Discussion

In this study, we investigated differences in American and Danish physicians' R/S characteristics and explored differences in their attitudes towards and practices of inquiring about R/S issues in the clinical encounter.

The items selected for the religiosity factor are based on the DUREL scale (Koenig and Büssing 2010). This scale is comprised of five items, covering three aspects of religiousness: organizational, non-organizational, and intrinsic or subjective religiosity. In the present study, we included the two measures of intrinsic religiosity ("To what extent do you agree with the following statement? I try hard to carry my religious beliefs over into all my other dealings in life" as well as "To what extent do you agree with the following statement? My whole approach to life is based on my religion."). Intrinsic religiosity represents the extent to which an individual sees his/her religiosity to be a central factor for meaning making. A third measure of intrinsic religiosity was included specifically for the practice of medicine, originally developed by Curlin et al. (6) that was obviously not in the original DUREL scale ("To what extent do you agree with the following statement? My religious beliefs influence my practice of medicine"). As regards organizational and non-organizational forms of religiosity, these were not included similarly in the Danish and American datasets and therefore could not be compared meaningfully. Instead, a fourth item was included ("To what extent do you consider yourself a religious person?"). This was done to capture forms

of religiosity which is not characterized by religious behaviors (or organization). Therefore, the factor we used to measure R/S characteristics does indeed measure several different things; however, this is a decision made from a theoretical and empirically corroborated standpoint that R/S is a multidimensional phenomenon (Koenig and Büssing 2010).

As expected, American physicians scored as more religious compared to Danish physicians. This finding corresponds with sociological findings about the religious characteristics of the two general populations. Although America and Denmark each have their unique and complex characteristics from historical, social and cultural developments, there are similarities, which seem to run across both countries, namely that they are both industrialized, democratic, and highly developed countries. In terms of the religion-cultural area, however, significant differences play out between Danish and US-American culture: Americans are, among other things, more religious, more accepting towards mixing religion and politics and attending religious services more often than northern Europeans (Andersen et al. 2008).

We found that compared to Danish physicians, American physicians deemed it more appropriate to discuss R/S issues in the clinical encounter when patients bring them up. If these attitudes translate into practice, those physicians deeming it inappropriate to discuss R/S issues could potentially disregard the core competency of patient-centred care (WHO 2005) with regards to communication in the clinical encounter—whereas the opposite could be true as well, namely that a physicians own religiosity could lead to an overemphasis on the patients religiosity. In order to investigate the relation between attitudes and practices, we therefore explored how attitudes—both among American and Danish physicians—towards discussing R/S issues if patients bring them up, related to the physicians' self-reported frequency of inquiring about R/S issues. This yielded a moderate correlation indicating some degree of relation between attitudes and self-reported practices. Therefore, it seems that, in practice, some physicians may give a lower priority to the core medical ambition of patient-centred care (WHO 2005) and thereby overlook potential significant personal factors in the patient that could contribute to both better or worse health.

Furthermore, comparing American and Danish physicians on the frequency with which they inquire about R/S issues across four clinical situations containing R/S topics, American physicians more often inquired about these topics. An explanation could be that in Denmark, religiosity is considered a primarily private matter (Rosen 2009), making Danish physicians more reluctant to inquire about these issues in general, regardless of religiosity. Another explanation could be that Danish physicians in general score lower on measures of religiosity compared to physicians of other nationalities, which has been found to be associated with a less frequent inquiry about R/S issues (Al-Yousefi 2012; Curlin et al. 2006; Lee et al. 2014; Lucchetti et al. 2016; van Randwijk et al. 2019) Therefore, we correlated R/S characteristics and frequency of inquiry separately for American and Danish physicians, in order to investigate whether religiosity was indeed related to frequency of inquiring about R/S issues for both groups. This revealed significant positive coefficients. Accordingly, it seems that religiosity is related to frequency of inquiring about R/S issues independently of national setting. Although the finding is not new, and maybe not surprising, it does contribute to the discussion about the dynamics between religiosity and clinical behavior and practice in Denmark and America. The fact that the inclination to inquire about R/S issues is lower in the Danish sample than in the American might not only reflect the privacy of religiosity. It could also indicate that Danish physicians are less religious in general, compared to American physicians. These explanations are not mutually exclusive, however. The influences of religiosity and national culture on physicians' integration of R/S into clinical practice ought to be explored further by employing qualitative research methods where tacit assumptions, intentions and ideas connected to the practice can be investigated more thoroughly.

Finally, when investigating the differences regarding American and Danish physicians' view on the appropriateness of the time they spend inquiring about R/S issues, American

physicians more often than Danish reported spending too little time. There are probably several plausible explanations for these results. As mentioned above, in Denmark religiosity is generally considered a private matter (Rosen 2009), potentially and paradoxically explaining both the lower frequency of inquiry into R/S issues of Danish physicians as well as their stronger tendency to think that the time they spend is the right amount, compared to American physicians. Another explanation could be that, in Europe, religion and science have traditionally been seen as two separate phenomena that should not partake in the same social spheres (Taylor 2007) whereas in America, most people do not think of science and religion as incompatible (Baker 2012).

Taken together, the findings indicate that national/cultural characteristics, such as the prevalence of religion, have a significant influence on communication in the clinical encounter. While it may not be surprising that physicians from countries that are generally more religious also turn out to be more religious than physicians from less religious countries, our findings do raise a number of pertinent concerns. From the view of a biomedical, predominantly evidence-based medical environment, valuing neutrality and objectivity, the clinical encounter ought not to be influenced by the values and orientations of the physician. However, from the perspective of patient-centered medical care, it is increasingly recognized that one of the factors that influence the clinical encounter and health outcomes is "the doctor as person" entailing the doctor's personal values and orientations (Mead and Bower 2000). This opens a potential problem. There is no way of telling if the influence of the doctor's subjectivity will be benign or the adverse in a particular clinical encounter, especially if the physician is not reflexive about his/her personal values and beliefs, and knowledgeable about the beliefs of patients and relatives.

This may be less of a problem in relatively homogenic cultures, since this study could indicate that, for instance, in a country with higher religiosity, you would also be more likely to encounter a religious physician and vice versa. On the other hand, this approach does not consider the experiences of minority groups, such as religious individuals in highly secular societies or non-religious individuals in highly religious societies. We believe this suggests it is particularly important to pay attention to these issues in countries with increasingly diverse demographics such as Europe and North America. Further studies into similarities and disparities between religious characteristics of physicians and general populations are needed to shed further light on these particular issues. Finally, one way to overcome problems regarding communication across cultural or religious diversities could be to strengthen and further develop research into, and curricula about, the influence of religion on communication in the clinical encounter.

*Limitations*

The study employed a large sample (N = 2053) which increase the generalizability, especially since physicians were selected on random from a stratified sample (Curlin et al. 2005) or entirely on random (van Randwijk et al. 2019). In some cases, all physicians from a speciality were included (van Randwijk et al. 2019). Admittedly, large sample sizes increase the risk of making Type 1 errors. However, we employed effect size measures in order to measure difference and practical significance instead of only statistical significance in order to compensate for this risk (Sullivan and Feinn 2012).

This study focused on religiosity, only. However, the Danish general population is known to be among the most secular in the world (Zuckerman 2008). Employing other measures of existential meaning-making, such as spirituality, may give better insights into the relations between physicians' beliefs and values and their medical practice, at least in Danish settings.

There was a 10-year time difference between the collection of the American and Danish data. Consequently, we cannot rule out that the differences identified between the nations could be consequences of time trends rather than of true variations in country specific characteristics.

The data are from 10 and 20 years ago which may cast an image of religiosity that does not correspond to the present. While we acknowledge this limitation, research also suggests that changes in religious values have not changed substantially over the past twenty years in both Denmark and the US which the World Values Survey, the European Values Study and the Pew Research Center support (Frederiksen and Gundelach 2019; Gundelach 2011; Serrán-Pagán y Fuentes 2020.)

There has been a slight change in the way people move away from organized religion towards individualistic spirituality, but the main traits of how many people believe in God, believe in life after death or go to church have been rather constant, and the trend internationally is that more favor than oppose an increased role for religion, especially in Africa, Asia and South America (Serrán-Pagán y Fuentes 2020).

Finally, with our study we are able to only humbly show how cultural and religious values are entangled. We do not propose any generalizable theory on religion and culture in medical practice but can only point to tendencies that merit further research. We are, however, concerned with tendencies favoring an ideal of "objective" medicine in which the impact of personal values is either ignored or even disregarded. See for this problem our recent article: "Religious Values in Clinical Practice are Here to Stay" (Kørup et al. 2016).

## 5. Conclusions

Knowledge of cultural differences in R/S characteristics, and their impact on attitudes and practices regarding R/S issues in the clinical encounter could prove useful in raising awareness about the issue, which in turn may aid health care professionals in ensuring that their practice stay patient centered. This seems especially relevant in increasingly ethnically and culturally diverse settings such as Denmark and America where a magnitude of adverse religious or non-religious affiliations flourishes. Further research into these issues could inform decision makers in formulating adequate strategies for interventions to further strengthen patient centered medical care. Moreover, knowledge about these issues could contribute to the process of producing medical curricula that aim at enhancing physicians' communicative and cultural competencies.

**Author Contributions:** Conceptualization, C.B.v.R. and N.C.H.; Funding acquisition, C.B.v.R. and N.C.H.; Investigation, C.B.v.R.; Methodology, C.B.v.R. and N.C.H.; Project administration, C.B.v.R. and N.C.H.; Resources, N.C.H.; Writing—original draft, C.B.v.R. and T.O.; Writing—review & editing, E.A.H., T.K.S., L.B., J.H., J.S. and N.C.H. All authors have read and agreed to the published version of the manuscript.

**Funding:** This research received no external funding.

**Institutional Review Board Statement:** Not applicable.

**Informed Consent Statement:** Not applicable.

**Data Availability Statement:** Not applicable.

**Conflicts of Interest:** The authors declare no conflict of interest.

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
