# Peer review of "Similarities and Differences between Danish and American Physicians’ Religious Characteristics and Clinical Communication: Two Cross-Sectional Surveys"

_religions, doi:10.3390/rel12020116_

Round 1

Reviewer 1 Report

The authors have examined an interesting question, that strengthens our understanding of the correlation between the level of religiosity of physicians and their inclinations toward spiritual care. The sample sizes in their studies are quite substantial, which strengthens the reliability of the study results.

I am concerned about the age of the data, however. The data used in this study is almost 20 and 10 years old, resp. How representative are the assessed attitudes of current attitudes of physicians in the USA and Denmark and, therefore, how informative is this study to understanding of spiritual care by physicians? If culture is such an important factor, as the authors seem to suggest, changes within the culture would impact the findings. My expectation would be that both cultures and their stances toward religion have changed significantly in the past decades? Please elaborate on this issue of time. 

Other questions:

What are the psychometric properties of the measure of R/S characteristics? At face value, it seems to assess (at least) two different things that, if separated might provide more information regarding the author's questions. The measure seems to assess: (a) Whether the physician considers religiosity to be important in their lives (b) whether the physician considers religious beliefs a 'private' or a 'public' matter. As the authors also indicate, in a society where religion is considered more private people might not score high on questions that might be read as meaning that one speaks about their own beliefs in their professional environments, even though they might be very religious. The very low scores of the Danish physicians on this measure would suggest the two go hand in hand, but I would like to have some more information on this to understand this question better.

Please also elaborate on the difference between the religiosity of physicians and the general population in both countries, preferably already in the introduction section. As indicated by the authors in the discussion section the low attention to religion might be appropriate to the cultural context. A mere statement that physicians are less religious than the general population does not convince me otherwise. How much less religious are they?

Though I am happy that the authors suggest a theoretical contribution of their study in the introduction section, I am struggling to understand it. The authors suggest on p.2, l. 46-49, that cross-cultural research would help to understand the influence of individual, cultural and religious characteristics on medical practice. However, these factors are not clearly disentangled in the present study and I wonder if they even can be disentangled. So how do the authors see this objective being realized in their study? And what kind of generalizable theory on religion and culture in medical practice do the authors have in mind?

Minor points:

Please describe the methods after the introduction and before the results, so it is clear how the results have come about and how they should be interpreted. 

Please use the same wording for the various variables in the results section and in the methods section. I got quite confused about which variables were being used for each of the described results.

Author Response

Dear Reviewer, 

Thank you for these valuable remarks and suggestions for improvement. 

We have addressed your suggestions in the manuscript as is visible also in the attached table with 1. Your suggestions, 2. Our comments to them, and 3. Our amendments in the Manuscript. 

Cordial greetings from all co-authors

Reviewer 2 Report

The study is fine but they are using data from the US that dates back 15-20 years. There has been much progress in spiritual care so if done today the data might be different. Also this study is strictly focused on religion. Most of the guidelines recommend addressing spiritual issues and specifically spiritual distress. The definitions  for spiritual and spiritual distress are broad and inclusive of secular and atheist. SO a more relevant study would be to ask about the broader issues of addressing spiritual as meaning and purpose and search for transcendence rather than "do you consider yourself a religious person".   I think while this data has merit and the study is solid I question the relevance with regard to clinical care.

Author Response

(The authors gave the same response as above.)

Round 2

Reviewer 1 Report

Thank you for addressing my comments and concerns in such a careful manner. 

Reviewer 2 Report

this is better they did focus on religion only vs spirituality more broadly defined   I am fine with the changes